# Effects of RU486 in Treatment of Traumatic Stress-Induced Glucocorticoid Dysregulation and Fear-Related Abnormalities: Early versus Late Intervention

**DOI:** 10.3390/ijms23105494

**Published:** 2022-05-14

**Authors:** Chen-Cheng Lin, Pao-Yun Cheng, Michael Hsiao, Yia-Ping Liu

**Affiliations:** 1Laboratory of Cognitive Neuroscience, Department of Physiology and Biophysics, Graduate Institute of Physiology, National Defense Medical Center, Taipei 11490, Taiwan; wsadhjkl@gmail.com; 2Genomics Research Center, Academia Sinica, Taipei 11529, Taiwan; mhsiao@gate.sinica.edu.tw; 3Department of Physiology and Biophysics, Graduate Institute of Physiology, National Defense Medical Center, Taipei 11490, Taiwan; pycheng@mail.ndmctsgh.edu.tw; 4Department of Psychiatry, Cheng Hsin General Hospital, Taipei 11220, Taiwan; 5Department of Psychiatry, Tri-Service General Hospital, Taipei 11490, Taiwan

**Keywords:** anxiety, fear memory, glucocorticoid receptor, hippocampus, posttraumatic stress disorder, time-dependent effect

## Abstract

Central glucocorticoid receptor (GR) activity is enhanced following traumatic events, playing a key role in the stress-related cognitive abnormalities of posttraumatic stress disorder (PTSD). GR antagonists are expected to have potential as pharmacological agents to treat PTSD-related symptoms such as anxiety and fear memory disruption. However, an incubation period is usually required and stress-induced abnormalities do not develop immediately following the trauma; thus, the optimal intervention timing should be considered. Single prolonged stress (SPS) was employed as a rodent PTSD model to examine the effects of early or late (1–7 versus 8–14 days after the SPS) sub-chronic RU486 (a GR antagonist) administration. Behaviorally, fear conditioning and anxiety behavior were assessed using the fear-conditioning test and elevated T-maze (ETM), respectively. Neurochemically, the expressions of GR, FK506-binding proteins 4 and 5 (FKBP4 and FKBP5), and early growth response-1 (Egr-1) were assessed in the hippocampus, medial prefrontal cortex (mPFC), amygdala, and hypothalamus, together with the level of plasma corticosterone. Early RU486 administration could inhibit SPS-induced behavioral abnormalities and glucocorticoid system dysregulation by reversing the SPS-induced fear extinction deficit, and preventing SPS-reduced plasma corticosterone levels and SPS-induced Egr-1 overexpression in the hippocampus. Early RU486 administration following SPS also increased the FKBP5 level in the hippocampus and hypothalamus. Finally, both early and late RU486 administration inhibited the elevated hippocampal FKBP4 level and hypothalamus GR level in the SPS rats. Early intervention with a GR antagonist aids in the correction of traumatic stress-induced fear and anxiety dysregulation.

## 1. Introduction

Posttraumatic stress disorder (PTSD) is a mental disorder involving fear memory abnormalities [1,2]. Individuals with the condition typically experience intrusive memories of the traumatic events [3,4]. The current treatment of PTSD is constrained by a shortage of integrated explanations of the therapeutic mechanisms applied to PTSD; some pharmacological treatments are effective against depressive and anxious symptoms, but not the core symptom of PTSD (i.e., over-retrieval of fear memory). For example, PTSD patients report dissatisfaction with selective serotonin reuptake inhibitor treatment because the vivid recollections of traumatic memories tend to remain [5,6]. Hence, exploiting new therapies that specifically target the pathophysiology of PTSD is critical.

Heightened central glucocorticoid receptor (GR) activity is an endocrinological characteristic of PTSD that causes elevated negative feedback on the hypothalamic–pituitary–adrenal (HPA) axis [2,7]. The overactive central GR serves as a key component in the development of PTSD, as it disrupts the regulation of fear memory [7,8]. Many clinical studies using functional magnetic resonance imaging (fMRI) and the Pavlovian fear conditioning test (PFCT) have demonstrated that some brain regions are specifically involved in the performances of fear conditioning and extinction [8,9]. These brain regions include the hippocampus [10,11,12,13], medial prefrontal cortex (mPFC) [14,15,16,17,18], and amygdala [10,12,19], and they are the so-called fear circuit [20,21]. Previous pre-clinical studies also have indicated that activation of the central GR in the fear circuit [22,23,24] may enhance fear memory and anxiety through cross-talk with the downstream signaling [15,25,26]. Thus, GR antagonists are potentially useful in the treatment of PTSD. However, since the development of PTSD proceeds along a progressively neuroplastic or adaptive course [16,17], the intervention time for GR antagonists is crucial, as GR might play different roles in the initial post-trauma period (when symptom onset has yet to occur) and the full-blown symptomatic stage [18,27].

Efforts have been made to investigate this timing effect using a GR antagonist, RU486 (or mifepristone) in animals following single-prolong stress (SPS), an animal model of the PTSD paradigm [28,29,30,31]. Ariki and colleagues used the contextual fear conditioning and extinction paradigm, and demonstrated that the administration of single-dose GR antagonist immediately after SPS could be useful in preventing the development of SPS-induced fear extinction memory impairment and hippocampal GR over-activation [32]. Ding and colleagues tried to investigate the effects of early versus late intervention using a three-day RU486 regime; they concluded that the effects of RU486 on the fear performance and central GR/FKBP5 mRNA expressions were independent of prior SPS [33,34]. However, we think that the issue of the optimal timing of RU486 intervention still needs to be clarified for the following reasons: (1) The SPS model they performed did not show the typical outcomes of fear extinction abnormality and anxiety symptoms. (2) Implementing an earlier and longer dosing regimens may make it easier to find the difference effects between early and late RU486 interventions. (3) Given that the role of GR involves interactions with the regulatory proteins of FKBP4/5 [35,36,37,38] and early growth response protein 1 (Egr-1) [25,39,40,41], we believe that it requires a comprehensive clarification of the changes in protein levels of GR/FKBP4/FKBP5/Egr-1 in the fear circuit.

The present study aimed to examine the effects of early and late RU486 regimen interventions on fear memory (using PFCT) and anxiety profile (using ETM) in rats after traumatic stress. Moreover, we examined whether PFCT was sensitive to the antagonism of GR receptors in a time-dependent manner after the traumatic stress. Thus, the RU486 regimens were introduced at two different times during this study, namely, early (1–7 days after SPS exposure) and late (8–14 days after SPS exposure) administration. Additionally, at the end of the study, plasma corticosterone, along with the expression of GR and the proteins related to its activation (i.e., FKBP4, FKBP5, and Egr-1) within stress-related brain areas, were measured. The results obtained from our study represent evidence justifying an appropriate timing for GR antagonist intervention in the development of PTSD.

## 2. Results

### 2.1. Three-Day Cue-Dependent Fear Conditioning Test

The data for freezing level exhibited no significant group-based effect and no interaction between group and trial on the first day (including during the pre-CS-1, CS habituation, and conditioning stages), the second day (including the pre-CS-2 and extinction stages), and the third day pre-CS-3 stage. ANOVA at the retrieval stage revealed a significant interaction between group and trial (F_(15,220)_ = 4.547, *p* < 0.001), which was driven by the differences between the CON-Veh/Veh and SPS-Veh/Veh groups in trials 1 (*p* < 0.001), 2 (*p* < 0.001), and 3 (*p* = 0.001); between the SPS-Veh/Veh and SPS-RU486/Veh groups in trials 1 (*p* < 0.001) and 2 (*p* < 0.001); and between the SPS-RU486/Veh and SPS-Veh/RU486 groups in trials 1 (*p* = 0.038) and 2 (*p* = 0.002). These results demonstrated that early RU486 intervention after SPS can adjust SPS-induced impairment of fear extinction retrieval ability (Figure 1).

### 2.2. ETM Test

For the three times of avoidance latency (i.e., baseline, avoidance 1 latency, and avoidance 2 latency), the data showed no significant effect at the baseline. ANOVA revealed a significant difference in the avoidance 1 latency among groups (F_(3,44)_ = 6.824, *p* = 0.001), and further analysis indicated significant differences between the CON-Veh/Veh and SPS-Veh/Veh (*p* = 0.034), and the SPS-RU486/Veh and SPS-Veh/RU486 (*p* = 0.009) groups, and there was a trend of difference between SPS-Veh/Veh and SPS-RU486/Veh (*p* = 0.059). ANOVA also exhibited a significant difference in the avoidance 2 latency among groups (F_(3,44)_ = 3.978, *p* = 0.014), and further analysis indicated no significant difference between groups. For the escape latency, the data analyses revealed no significant between-group discrepancies. These results indicate that early RU486 intervention after SPS had a tendency to reduce the SPS-increased conditioned anxiety (i.e., SPS-increased avoidance 1 latency) (Figure 2).

### 2.3. GR, FKBP4, FKBP5, and Egr-1 Expression in the Hippocampus

ANOVA revealed a significant difference in the hippocampus GR expression among groups (F_(3,20)_ = 4.649, *p* = 0.013). The post hoc Bonferroni test suggested a trend of difference between the CON-Veh/Veh and SPS-Veh/Veh (*p* = 0.063) groups. ANOVA also demonstrated a significant difference in the hippocampus FKBP4 expression among groups (F_(3,20)_ = 4.853, *p* = 0.011), and further analyses revealed significant differences between the CON-Veh/Veh and SPS-Veh/Veh (*p* = 0.025), the SPS-Veh/Veh and SPS-RU486/Veh (*p* = 0.038), and the SPS-Veh/Veh and SPS-Veh/RU486 (*p* = 0.039) groups. For FKBP5 expression in the hippocampus, the data showed a significant difference among groups (F_(3,20)_ = 6.018, *p* = 0.004), and further analyses showed significant difference between the SPS-Veh/Veh and SPS-RU486/Veh (*p* = 0.026) groups. The data showed significant differences in terms of Egr-1 expression in the hippocampus among groups (F_(3,20)_ = 17.83, *p* < 0.001), which was driven by the significant differences between the CON-Veh/Veh and SPS-Veh/Veh (*p* < 0.001), the SPS-Veh/Veh and SPS-RU486/Veh (*p* < 0.001), and the SPS-RU486/Veh and SPS-Veh/RU486 (*p* = 0.05) groups. These results demonstrate that early RU486 intervention after SPS averted the SPS-increased levels of FKBP4 and Egr-1. Besides, early RU486 intervention after SPS increased FKBP5 expression in the hippocampus, and late RU486 intervention after SPS also reversed the SPS-increased FKBP4 level (Figure 3).

### 2.4. GR, FKBP4, FKBP5, and Egr-1 Expression in mPFC

No significant differences in the expression of GR (F_(3,20)_ = 0.694, *p* = 0.566) or FKBP4 (F_(3,20)_ = 0.338, *p* = 0.798) in the mPFC were observed among groups. However, for FKBP5 expression in the mPFC, ANOVA revealed a significant difference among groups (F_(3,20)_ = 3.81, *p* = 0.026). A post hoc Bonferroni test indicated a significant difference between CON-Veh/Veh and SPS-RU486/Veh groups (*p* = 0.034). ANOVA also showed a significant difference in terms of Egr-1 expression in the mPFC among groups (F_(3,20)_ = 3.18, *p* = 0.046), but the post hoc Bonferroni test exhibited no significant difference among the groups. These results indicated that SPS and RU486 treatment had no effect on the expressions of GR, FKBP4, FKBP5, and Egr-1 in the mPFC (Figure 4).

### 2.5. GR, FKBP4, FKBP5, and Egr-1 Expression in the Amygdala

No significant differences in terms of the expression of GR (F_(3,20)_ = 0.264, *p* = 0.85), FKBP4 (F_(3,20)_ = 0.512, *p* = 0.679), FKBP5 (F_(3,20)_ = 0.491, *p* = 0.692), and Egr-1 (F_(3,20)_ = 0.119, *p* = 0.948) in the amygdala were observed among groups. These results indicated that SPS and RU486 exerted no effect on the expression of GR, FKBP4, FKBP5, and Egr-1 in the amygdala (Figure 5).

### 2.6. Plasma Corticosterone Level

The data on plasma corticosterone level revealed significant differences among groups (F_(3,28)_ = 5.472, *p* = 0.004), which was driven by the differences between the CON-Veh/Veh and SPS-Veh/Veh (*p* = 0.013) and the SPS-Veh/Veh and SPS-RU486/Veh (*p* = 0.025) groups. These results demonstrated that early RU486 intervention after SPS prevented SPS-reduced plasma corticosterone (Figure 6A).

### 2.7. GR, FKBP4, and FKBP5 Expression in the Hypothalamus

In terms of GR expression in the hypothalamus, the data showed significant differences among groups (F_(3,20)_ = 5.621, *p* = 0.006), with further analyses suggesting significant differences between the CON-Veh/Veh and SPS-Veh/Veh (*p* = 0.032), the SPS-Veh/Veh and SPS-RU486/Veh (*p* = 0.023), and the SPS-Veh/Veh and SPS-Veh/RU486 (*p* = 0.011) groups. ANOVA showed a significant difference in the hypothalamus FKBP4 expression among groups (F_(3,20)_ = 5.621, *p* = 0.027), with a trend of differences between the CON-Veh/Veh and SPS-Veh/Veh (*p* = 0.053) and the SPS-Veh/Veh and SPS-RU486/Veh (*p* = 0.050) groups. ANOVA also revealed significant differences among groups (F_(3,20)_ = 6.718, *p* = 0.003), with further analysis showing significant differences between the CON-Veh/Veh and SPS-RU486/Veh (*p* = 0.004), the SPS-Veh/Veh and SPS-RU486/Veh (*p* = 0.011), and the SPS-RU486/Veh and SPS-Veh/RU486 (*p* = 0.034) groups. These results demonstrated that both early and late RU486 intervention after SPS reversed the SPS-induced GR overexpression, and early RU486 intervention had a tendency to reduce the SPS-increased expression of FKBP4 in the hypothalamus. In addition, early RU486 intervention after SPS increased FKBP5 expression in the hypothalamus (Figure 6B–D).

## 3. Discussion

Individuals may go through an incubating period following exposure to traumatic stress and their fear and anxiety symptoms arise weeks or months later. Traumatic stress-induced GR overactivity within the stress circuit is closely associated with this symptom development, and the selective GR antagonist RU486 aids in the inhibition of central GR activity. However, the most appropriate timing for RU486 intervention still requires clarification. Our data demonstrated that, distinct from late RU486 administration, early RU486 administration exhibited beneficial behavioral and neurochemical effects on SPS rats. Behaviorally, early RU486 administration improved fear extinction retrieval, and had a tendency to reduce conditioned anxiety. Neurochemically, early RU486 administration prevented the SPS-induced overexpression of hippocampal Egr-1 and reversed the SPS-induced reduction in plasma corticosterone levels. In addition, early RU486 administration also enhanced the FKBP5 levels in the hippocampus and hypothalamus in SPS rats. In addition, both early and late RU486 administration reduced the overexpression of hippocampal FKBP4 level and hypothalamus GR level after SPS. These findings help elucidate the role of GR in the development of fear memory dysregulation following traumatic events and provide new insights into the therapeutic potential of early RU486 administration in the treatment of PTSD. These findings are further discussed below.

In the present study, the SPS model that we implemented showed the expected behavioral outcomes of fear memory dysregulation and anxiety symptoms [42,43]. For the fear memory dysregulation, the SPS rats exhibited impairment in terms of fear extinction retrieval, despite having learned that the fear conditioning should be eliminated; this finding is consistent with the previous studies, including research on clinical patients with PTSD [7,44]. We suggest that the impairment of fear extinction retrieval may originate from disruption of the consolidation of fear extinction memory, rather than an inability to learn the fear extinction.

For the anxiety symptoms, our ETM results show that SPS lengthened the avoidance 1 latency but not that of escape, with SPS rats preferring to spend more time in secure or familiar places before visiting dangerous or unfamiliar ones. This suggests that previous distressing events increased rats’ hesitancy, a form of anxiety reflecting ambivalence. However, there were no differences among the groups in the ETM baseline and avoidance 2 latency, which should be attributed to the rat’s instinctive escape from the operator and the ceiling effect, respectively. According to our expectation, whether the SPS-induced abnormalities can be corrected is highly dependent on the timing of GR activity intervention. We found that only early RU486 administration reduced the freezing level in the retrieval stage (in fear conditioning tests) and had a tendency to increase the avoidance latency (in ETM test) in the SPS rats. These results demonstrate that the incubating period following exposure to traumatic stress is a critical time for the GR-associated mechanisms to exert their effects.

The stage-dependent effects of RU486 can also be applied to explain the changes in the expression of GR-related proteins in the brain areas processing fear memory, despite disparities in effects among the hippocampus, mPFC, and amygdala. The hippocampus was the most consistent, in which SPS increased the expression of Egr-1 and early RU486 intervention reversed the effect. Previous studies have indicated that elevated hippocampal Egr-1 contributes to fear memory consolidation [25,26,45]. Interestingly, the GR-related profile in the mPFC and amygdala appears to be less disturbed in the present and previous studies [20,46,47], and there is no timing effect of RU486 in the present studies. Notably, GR expression in the mPFC and amygdala are not as abundant as they are in the hippocampus, potentially explaining the insensitivity of mPFC and amygdala GR to SPS and RU486 [48]. It is also consistent with the fact that GR is not the dominant factor in the regulation of mPFC and amygdala-related fear and anxiety in PTSD pathophysiology; other neural substrates, including serotonin and dopamine, should also be considered [49,50]. Taken together, we suggest that the hippocampal GR-dependent Egr-1 level may serve as an index for both PTSD pathogenesis and treatment effects [15,25,26], which explains why early RU486 administration moderated the SPS-induced disruption of fear extinction memories in the present study.

The HPA axis is involved in the pathogenesis of PTSD, as it exerts a negative feedback to regulate corticoid functions [2,51]. In the present study, SPS reduced the peripheral corticosterone level and increased GR expression in the hypothalamus; only early RU486 administration could adjust the SPS-induced GR overactivity and HPA axis dysfunction. Restoring plasma cortisol levels following the blockade of GR receptors may be beneficial for PTSD patients. In addition to the physiological benefits, such as improved immunity and reduced inflammation, clinical studies of low-dose glucocorticoid administration have shown a significant reduction in PTSD symptoms, which may be attributable to the effects of cortisol, strengthening fear extinction by adjusting GR and FKBP5 functions in stress circuits [52,53]. These results may explain the paradox where both GR agonists and antagonists are useful in treating PTSD. The two might contribute differently at different stages of the development of fear memory dysregulation; GR antagonists inhibit fear memory consolidation [26,54], whereas GR agonists improve fear extinction [53,55].

It is notable that, in the present study, FKBP5 expression was significantly increased in the hippocampus and hypothalamus in rats that received early RU486 administration following SPS exposure. Studies have revealed that a higher FKBP5 performance yields lower GR sensitivity [56,57]. Thus, early RU486 administration likely prevented excessive GR-related transcription through the upregulation of FKBP5 activity after SPS exposure. By contrast, this situation did not occur in the late RU486 group. The time-dependent effect of RU486 on FKBP5 in SPS rats suggests that the early stage after traumatic stress is more responsible for addressing the GR dysregulation induced by such stress.

The present study entailed certain notable limitations, emphasizing the need for caution when interpreting the study’s findings. First, the expression of the upstream GR and downstream Egr-1 was sometimes inconsistent (e.g., in the hippocampus), which may have been affected by (1) the interaction between FKBP4 and FKBP5 or (2) another glucocorticoid-targeting receptor, the mineralocorticoid receptor [51,58]. Second, the effects of early or late RU486 administration were obtained from a single dose of a sub-chronic regimen, and the use of multiple doses of RU486 needs to be assessed in future studies. Third, the present study did not investigate the effect of RU486 on control animals, so the results obtained from the present study were only suitable for interpreting SPS rats (or patients diagnosed with PTSD), but not for the control rats (or healthy population). Fourth, given that ventral and dorsal subregions of hippocampus have different functions [59], the use of whole hippocampus in the present study may cause some potential effects that cannot be found.

In summary, in the present study, we compared the effects of early and late RU486 interventions on traumatic stress-induced behavioral abnormalities and GR-related profile. In behavioral terms, only early RU486 intervention after SPS exposure alleviated the fear extinction deficit and had a tendency to inhibit SPS-induced conditioned anxiety. In terms of endocrinology, only early RU486 intervention restored the SPS-reduced corticosterone plasma levels. For the neurochemical profile, the hippocampus reflected GR activity and the downstream profiles of FKBP4 and Egr-1 with high sensitivity following SPS exposure; it was also an accurate measure of RU486 efficacy in the treatment of SPS-induced dysfunctions. Our results validate the utility of early RU486 intervention and the theory of GR activity’s influence on the development of PTSD.

## 4. Materials and Methods

### 4.1. Animals

A total of 48 male, 8-week-old Sprague–Dawley rats (BioLASCO, Taipei, Taiwan Co., Ltd., Taipei, Taiwan) were used in this study. The experimental design is illustrated in the Figure 7. All rats were randomly assigned to one of four groups (each *n* = 12), namely, the control vehicle (CON-Veh)/Veh, SPS-Veh/Veh, SPS-RU486/Veh (early RU486 administration, days 2–8), and SPS-Veh/RU486 (late RU486 administration, days 9–15). All 12 rats in each group were used for behavioral tests, and 6 rats and 8 rats from each group were randomly selected for the Western blot and plasma corticosterone level assay, respectively. The experimental design of this study is illustrated in Figure 1. All rats (2 rats/cage) were housed in a holding facility with controlled temperature (25 °C ± 1 °C), controlled humidity (50% ± 10%), and 12-h light–dark cycles (lights on from 07:00–19:00). All animals received a standard laboratory chow diet (Ralston Purina, St. Louis, MO, USA) and sterile water ad libitum. All experiments were conducted with approval from the NDMC animal care committee (IACUC-20-308, approved on 10 September 2020), and substantial efforts were exerted to ensure that the number of animals used was minimal and their suffering during the experiments was minimized. All experiments performed were verified in accordance with the relevant guidelines and regulations of Taiwan.

### 4.2. SPS

The SPS procedure followed a similar protocol to that described in previous works [29,60,61] but without the undisturbed period of 7 days. In brief, each rat was restrained in an animal holder for 2 h, compelled to complete a 20 min swim, and then, after a 15 min recovery, was exposed to approximately 5% ethyl vapor until loss of consciousness (defined as loss of pinch reflex in the extremities), whereupon each rat was returned to its home cage. After a 24 h quiescent period, all rats were administered RU486 or vehicle for 14 days in their home cages. Notably, the CON group rats remained in their home cages during the period in which the SPS group was undergoing the SPS procedure.

### 4.3. Drugs

RU486 (Tocris Bioscience, Bristol, Avon, UK), a GR antagonist, was dissolved in propylene glycol (Veh, Sigma-Aldrich, St. Louis, MO, USA). All rats were intraperitoneally (i.p.) injected with RU486 (20 mg/kg) or propylene glycol [62,63] once daily for 14 days in their home cages. The doses of RU486 were chosen based on previous studies [64,65]. The SPS-RU486/Veh group rats were administered a 7 day RU486 regimen from days 2–8, and then propylene glycol from days 9–15. The SPS-Veh/RU486 group rats were initially administered 7 days of propylene glycol, and then 7 days of RU486. To exclude the potential toxic effects of propylene glycol [66], the CON-Veh/Veh and SPS-Veh/Veh group rats received propylene glycol for 14 days after SPS exposure (days 2–15). The drugs were freshly prepared to produce a total injection volume of 1.0 mL/kg body weight.

### 4.4. Three-Day Cue-Dependent Fear Conditioning Test

In this study, a 3-day cue-dependent fear conditioning test was performed on days 17–19, with a similar protocol to that employed in previous studies [60,67,68,69,70]. In brief, the operant chamber (chamber size: 30 × 24 × 26 cm^3^) used in this test was constructed of Plexiglas and aluminum (Clever Sys., Inc., Reston, VA, USA). The chamber was placed in a wooden box to attenuate noise. A house light (75 W white light) and a fan (65 dB, as background noise) were mounted on the wall inside the wooden box. A total of 18 stainless steel rods were set on the floor (5 mm in diameter and spaced 1.5 cm from each other) of the chamber to supply an electrical foot shock.

During the test, an 8 W light-emitting diode bulb light (10 s, 60 s inter-trial interval) was used as the conditioned stimulus (CS), and a 1 mA electrical foot shock (1 s) was used as the unconditioned stimulus (US). Two different contexts (context 1 and context 2) were employed to reduce interference from the chamber environment. Context 1 was performed for the first day and employed the white-colored house light, fan operation, and 1% acetic acid odor; context 2 was carried out on the second and third days and involved a red-colored house light, no fan operation, and the introduction of 1% ammonia odor. Contexts 1 and 2 were employed in the same operant chamber.

Several stages were involved in this 3-day protocol. On the first day, the rats were placed in the chamber (context 1) without the CS or US for 3 min (pre-CS-1 stage) and then received 5 CSs (CS habituation stage) and 7 CSs that co-terminated with US for conditioning (conditioning stage). On the second day, the rats were returned to the same chamber (context 2) without CS and US for 3 min (pre-CS-2 stage) and then received 15 CSs for extinction learning (extinction stage). On the final day, the rats were returned to the chamber (context 2) without CS and US for 3 min (pre-CS-3 stage) and then received 6 CSs for extinction retrieval (retrieval stage). During this test, freezing behavior in the rats was recorded every 60 s by a FreezeScan system (Clever Sys., Inc., Reston, VA, USA). The rat received US and CS an appropriate number of times, which was obtained after internal testing with good fear conditioning, fear elimination, and recall elimination without causing additional stress to the rat.

### 4.5. ETM

The ETM was employed on days 21 and 22, with a similar protocol to that employed in our previous study [60,71,72]. ETM’s avoidance and escape latencies reflect generalized and panic anxiety, respectively [72,73,74]. The ETM apparatus was elevated 50 cm above the floor and had three arms of equal dimensions (50 × 12 cm^2^), with one enclosed arm surrounded by a 40 cm-high wall, and two open arms with 1 cm-high Plexiglas rims. The two open arms were perpendicular to the enclosed arm.

For the ETM training, rats were pre-exposed to the open arm for 30 min, with the enclosed arm blocked during the training. After 24 h, the rats were placed at the end of the enclosed arm three times and the time required to leave the arm using all four paws was recorded and used as the baseline, avoidance 1 latency, and avoidance 2 latency, wherein the time required to leave the arm by using all four paws was recorded. After the avoidance 2 stage, the rat was placed at the end of the open arm and the time spent entering the enclosed arm was recorded (escape latency), the inter-trial interval was 60 s, and the time for each trial lasted up to 300 s. Once the rats exited the enclosed arm (baseline and avoidance latency) or entered the enclosed arm (for escape latency), the trial was stopped and the rat was quickly placed into their home cage to inhibit the possibility of habituation effects. The degrees of generalized and panic anxiety were examined based on the latencies of the ETM avoidance and escape phases, respectively.

### 4.6. Animal Euthanasia

Animal sacrifice was performed at the end of experiment (i.e., 24 h after the ETM test); all rats were placed in a closed Plexiglas chamber (chamber size: 30 × 15 × 15 cm^3^), and the rats were decapitated after inhaling 5% isoflurane vapor until they died. The above sacrifice process was executed one rat at a time.

### 4.7. Plasma Corticosterone Level

Blood samples were collected in a tube containing 1 μL of heparin sodium and mixed immediately after euthanasia (16:00–18:00), and then the blood samples were centrifuged at 6000× *g* at 4°C for 10 min. The supernatants were used to assay the plasma corticosterone concentration using a corticosterone EIA Kit (Cayman Chemical Company, Ann Arbor, MI, USA) with the absorbance at 405 nm.

### 4.8. Western Blotting

After euthanasia, the target brain areas (hippocampus, mPFC, amygdala, and hypothalamus) were quickly and carefully dissected on an ice-cold plate in accordance with the method of Paxinos and Watson [75]. The brain areas were homogenized in the optimal volume of cold lysis buffer (0.15 M NaCl, 20 mM Tris (pH 8.0), 10 mM Na_4_P_2_O_7_, 1 mM EDTA, 1% NP40, and 5% glycine), containing a protease inhibitor (Roche, Mannheim, Germany) and a phosphatase inhibitor cocktail (Veh, Sigma-Aldrich, St. Louis, MO, USA). After being centrifuged (14,000× *g*, 4 °C for 30 min), the supernatant was collected and equal amounts of protein (50 μg, determined based on the protein assay dye manual (Bio-Rad Laboratories, Hercules, CA, USA)) were denatured by heating at 95 °C for 10 min and then separated using SDS-polyacrylamide gel electrophoresis with 10% polyacrylamide gels. Separated proteins in the gel were then electroblotted onto polyvinylidene difluoride membranes (Bio-Rad, Alfred Nobel Dr, Hercules, CA, USA) for Western blot analysis. Membranes were blocked with 5% bovine serum albumin in TBST (Tris-buffered saline, 0.1% Tween 20) (136.8 mM NaCl, 24.7 mM Tris base, 2.68 mM KCl, and 0.1% Tween 20 (pH: 7.4)) at room temperature for 1 h. The primary antibodies of GR (Cat. No.: Ab109022, Abcam, Cambridge, UK), FKBP4 (Cat. No.: Ab129097; Abcam, Cambridge, UK), FKBP5 (Cat. No.: 4154S; Cell Signaling Technology, Danvers, MA, USA), Egr-1, and GAPDH (Cat. No.: MAB374; Merck Millipore, Burlington, MA, USA) were used with dilutions of between 1:1000 and 1:5000 for reaction overnight at 4 °C. Thereafter, the corresponding secondary antibody was used with 1:5000 dilution at room temperature for 1 h. The blots were further washed and the immunoreactive bands were detected using Western Chemiluminescent HRP substrate (Cat. No.: WBKLS0500; Merck Millipore, Burlington, MA, USA) and recorded using a UVP BioSpectrum 500 Imaging System (UVP, LLC, Upland, CA, USA). In addition, we calibrated the concentration of antibodies using the standardized method: the immunoreactive bands were detected and the corresponding optical density (OD) values were measured, with each OD value divided by the average OD values of the corresponding CON-Veh/Veh group to obtain standardized OD values. The data were calculated using the following formula: standardized target protein OD value ÷ corresponding standardized GAPDH OD value.

### 4.9. Data Analyses

In the present study, one-way analysis of variance (ANOVA) with a between-subjects factor of group (4 groups: CON-Veh/Veh, SPS-Veh/Veh, SPS-RU486/Veh, and SPS-Veh/RU486) was performed on the data of the ETM test, Western blot, and plasma corticosterone level. Two-way repeated measures ANOVA with a between-subjects factor of group and a within-subjects factor of trial were performed on the freezing level of the fear-conditioning test. The statistically significant main effects were subjected to post hoc comparison using the Bonferroni post-hoc test; if interactions were found, the data were split for the simple main effect. In the present study, the statistical importance only exhibited the difference in 2 aspects: (i) the different between rats with SPS and rats without SPS (i.e., CON-Veh/Veh versus SPS-Veh/Veh), and (ii) the different between timing effects of RU486 intervention in SPS rats (SPS-Veh/Veh versus SPS-RU486/Veh versus SPS-Veh/RU486). *p-*values of <0.05 were defined as statistically significant, and *p-*values between 0.05 and 0.07 were defined as trending. All statistical analyses in the present study were employed by SPSS 16.0 for Windows software (Chicago, IL, USA).

## Figures and Tables

**Figure 1 ijms-23-05494-f001:**
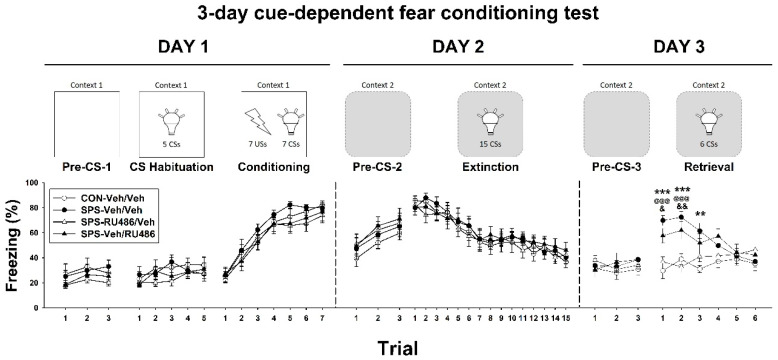
Three-day cue-dependent fear-conditioning test after SPS paradigm and early/late RU486 intervention. The data represent the mean ± SEM. *n* = 12 for each group. Statistical analysis was employed with a two-way repeated measures ANOVA followed by Bonferroni post hoc testing. ** *p* < 0.01, *** *p* < 0.001, CON-Veh/Veh vs. SPS-Veh/Veh; @@@ *p* < 0.001, SPS-Veh/Veh vs. SPS-RU486/Veh; & *p* < 0.05, && *p* < 0.01, SPS-RU486/Veh vs. SPS-Veh/RU486. CS: conditioned stimulus; US: unconditioned stimulus; context 1: the chamber with white house light, fan on and full of 1% acetic acid smell; context 2: the chamber with red house light, fan off and full of 1% ammonia smell. The contexts 1 and 2 were employed in the same operant chamber.

**Figure 2 ijms-23-05494-f002:**
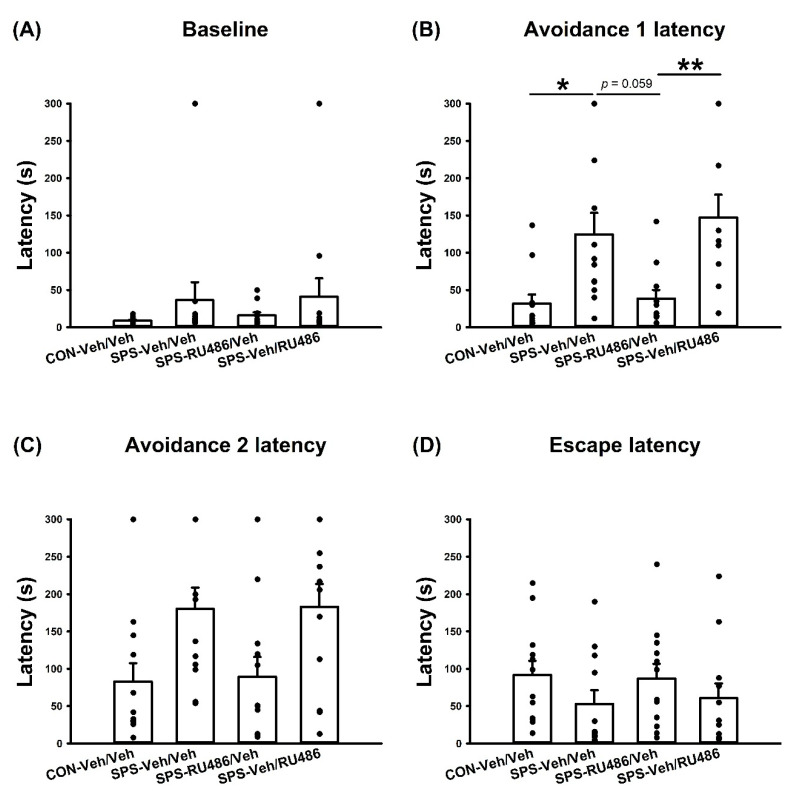
The preferences of the (**A**) Baseline (**B**) Avoidance 1 latency (**C**) Avoidance 2 latency and (**D**) Escape latency of ETM test after SPS paradigm and early/late RU486 intervention. The data represent the mean ± SEM. *n* = 12 for each group. Statistical analysis was employed with a one-way ANOVA followed by Bonferroni post hoc testing. * *p* < 0.05, ** *p* < 0.01.

**Figure 3 ijms-23-05494-f003:**
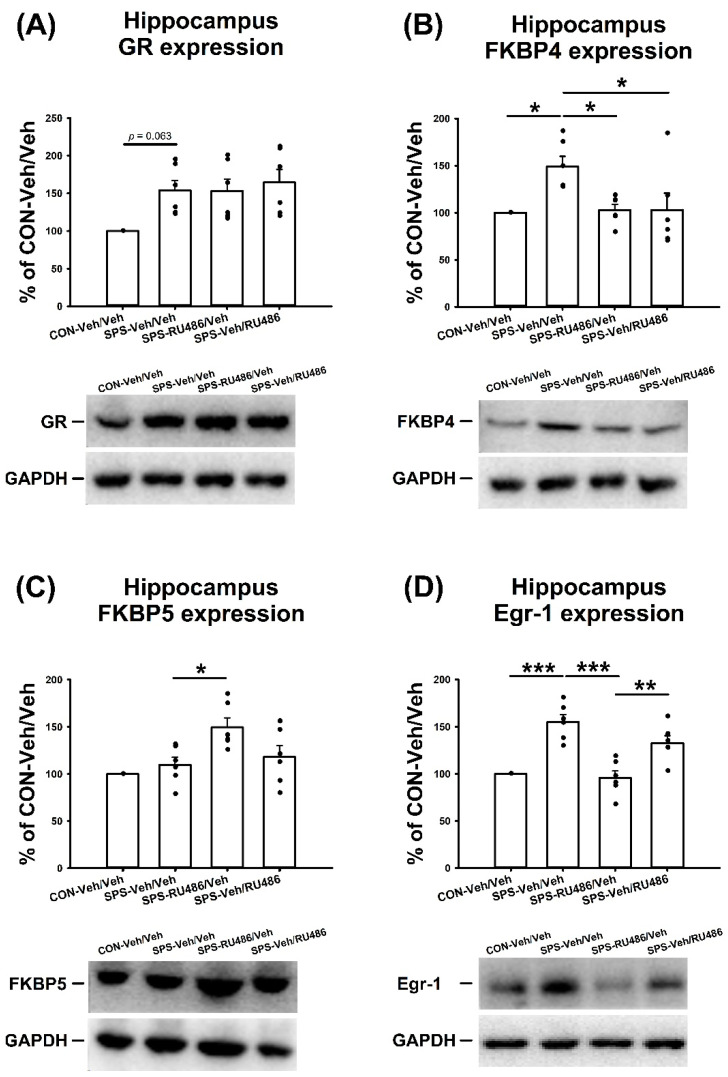
The expressions of the (**A**) GR, (**B**) FKBP4, (**C**) FKBP5, and (**D**) Egr-1 in the hippocampus after SPS paradigm and early/late RU486 intervention. The data represent the mean ± SEM. *n* = 6 for each group. Statistical analysis was employed with a one-way ANOVA followed by Bonferroni post hoc testing. * *p* < 0.05; ** *p* < 0.01; *** *p* < 0.001.

**Figure 4 ijms-23-05494-f004:**
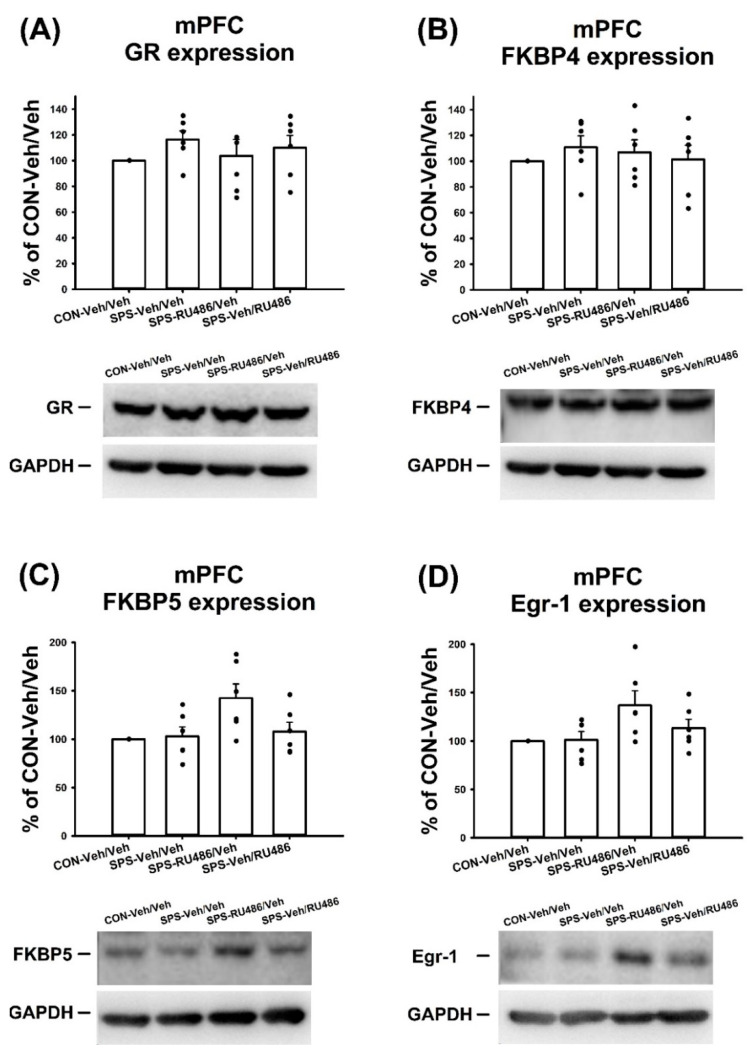
The expressions of the (**A**) GR, (**B**) FKBP4, (**C**) FKBP5, and (**D**) Egr-1 in the mPFC after SPS paradigm and early/late RU486 intervention. The data represent the mean ± SEM. *n* = 6 for each group. Statistical analysis was employed with a one-way ANOVA followed by Bonferroni post hoc testing.

**Figure 5 ijms-23-05494-f005:**
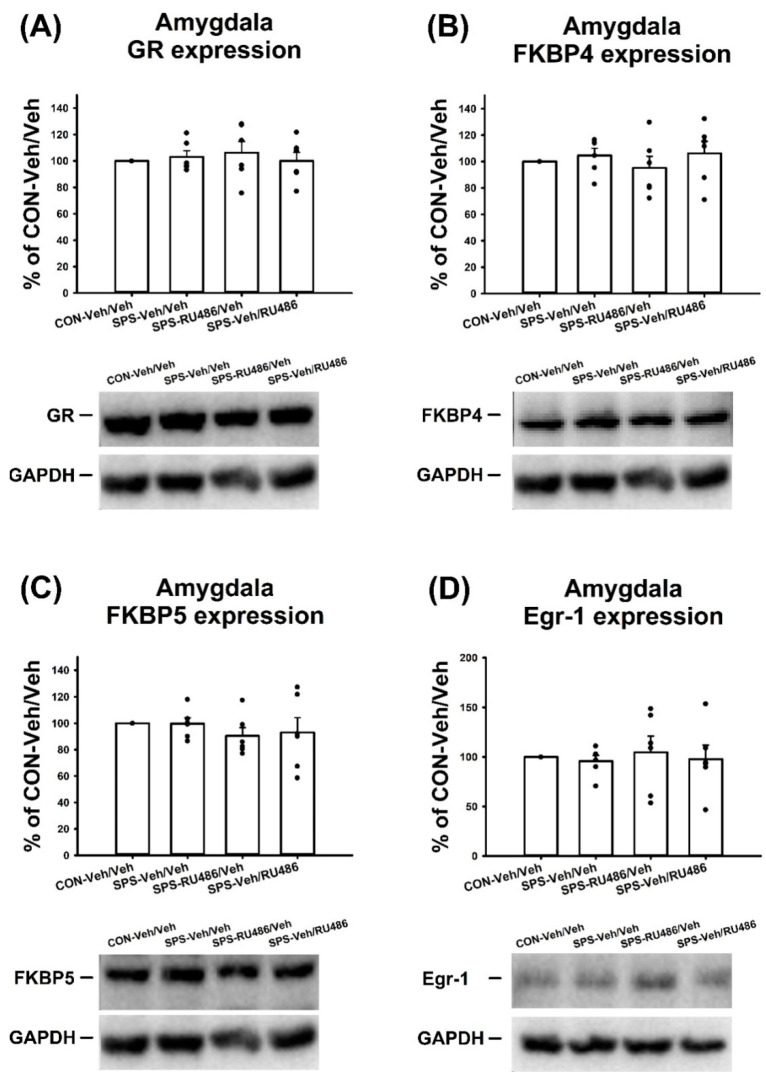
The expressions of the (**A**) GR, (**B**) FKBP4, (**C**) FKBP5, and (**D**) Egr-1 in the amygdala after SPS paradigm and early/late RU486 intervention. The data represent the mean ± SEM. *n* = 6 for each group. Statistical analysis was employed with a one-way ANOVA followed by Bonferroni post hoc testing.

**Figure 6 ijms-23-05494-f006:**
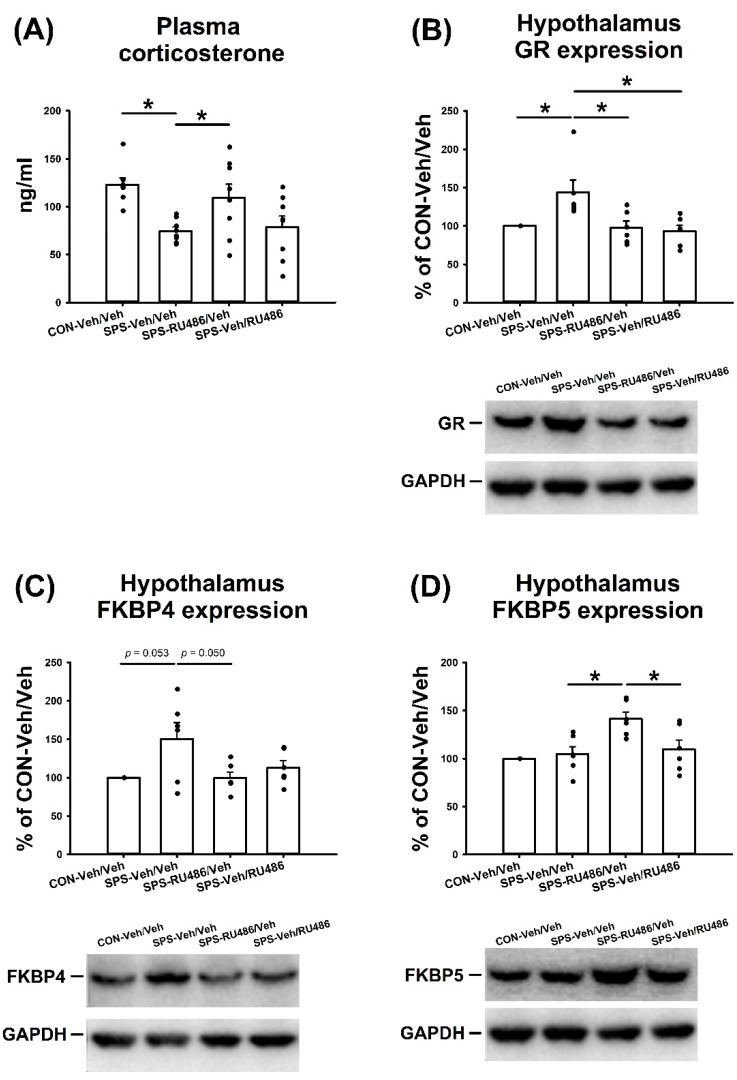
(**A**) The concentration of Plasma corticosterone concentration and the expressions of the (**B**) GR, (**C**) FKBP4, (**D**) FKBP5 in the hippocampus after SPS paradigm and early/late RU486 intervention. The data represent the mean ± SEM. *n* = 8 in the data of plasma corticosterone level, and *n* = 6 in the data of western blot for each group. Statistical analysis was employed with a one-way ANOVA followed by Bonferroni post hoc testing. * *p* < 0.05.

**Figure 7 ijms-23-05494-f007:**
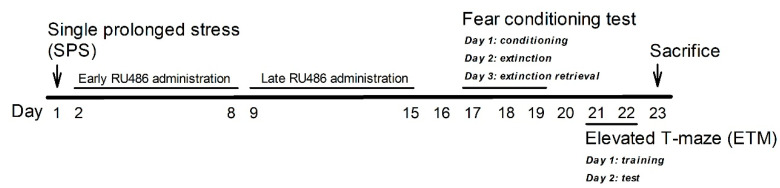
Schematic illustration of the experimental design in the present study.

## Data Availability

The datasets generated and analyzed in the present study are available from the corresponding author on reasonable request.

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
