# Peer review of "Effects of RU486 in Treatment of Traumatic Stress-Induced Glucocorticoid Dysregulation and Fear-Related Abnormalities: Early versus Late Intervention"

_ijms, 2022, doi:10.3390/ijms23105494_

Round 1
Reviewer 1 Report
The manuscript reports attempting to characterize the effect of GR antagonist as potential treatment for PTDS-related symptoms. Despite the potential of the study, authors should consider to treat also control animals. This makes the data difficult to interpret since they cannot really describe the effects obtained as “preventive”.
Why authors changed the number of US presented in the three days of the fear conditioning test?
How long after the last behavioral test the animals were sacrificed? Please add this information in the method section.
Which is the effect of the treatment in control animals? Even if from a translational point of view it is not useful to treat “healthy” subjects, how authors can compare and so comment the data obtained by comparing the SPS animals treated with CON animals untreated?
Why authors used the one way anova? They should use the two-way anova since the factors are the procedure and the treatment, so as consequence, all experimental groups should be exposed to both the variables. To state that the treatment has a real effect in SPS animals, authors should treat control rats and study the effect of the only treatment.
For example, figure 5C: how authors can say that “early RU486 intervention after SPS increases FKBP5 (and also Egr-1, that is not significative p=0.059) in the mPFC”? The increase in FKBP5 in SPS-RU486/Veh could be due to the only treatment with RU486 (but it is impossible to evaluate this hypothesis if they don’t have CON-RU486 group). Indeed, the SPS-Veh/Veh and SPS-RU486/Veh are not statistically different ( p=0.054 for FKBP5 and p= 0.071 for Egr-1).
Another example in figure 4A: since the SPS-veh/veh group induced per se an increased expression of GR independently from the treatment, authors cannot compare SPS-RU486/Veh and SPS-Veh /V RU486 with the CON/Veh/Veh since these experimental groups are characterized by different procedure (CON-SPS) and treatment (early-late)
In which subcellular fraction the protein levels have been measured? The subcellular fractionation and the levels of GR in the fraction, as well as the level of FKBP5 in the cytosol could lead information about GR translocation in the nucleus or its retention in the cytosol.
Have the authors analyzed the whole hippocampus? The dissection of ventral and dorsal subregions could help the author to better interpret the effects found.
Authors should include the meaning of the results not in the figure legend but in the text. Please move the summary sentence at the end of each paragraph.
Minor:
Please increase the quality of the figure 2. Some SEM are missing.
In the supplementary, please add the experimental groups above the WB bands.
Author Response
Dear reviewer 1, April 30, 2022
Thanks for your comments on IJMS-1663969, and we realize that a lot of efforts you have made to make this manuscript more readable and concise, so it may provide greater scientific contribution to the research feld of fear memory. Folloiwng your suggestions we revised the manuscript fully and also received a carefully English Edition work from MDPI before its submission. Please see our replies to your comment point by point as the following.
Best Regards,
Liu, Yia-Ping M.D., Ph.D
Professor, Department of psychiatry, Cheng Hsin General Hospital, Taipei, Taiwan Director, Laboratory of cognitive neuroscience, Department of physiology, National Defense Medical Center, Taipei, Taiwan
The manuscript reports attempting to characterize the effect of GR antagonist as potential treatment for PTDS-related symptoms. Despite the potential of the study, authors should consider to treat also control animals. This makes the data difficult to interpret since they cannot really describe the effects obtained as “preventive”.
Why authors changed the number of US presented in the three days of the fear conditioning test?
Reply: Due to the purpose of Pavlovian fear condition paradigm that the degree/amount of fear reaction needs to be examined in three different stages (namely conditioned stage, extinction stage, and retrieval stage), thus the strength of fear corresponding signals and the course of paring for learning process in animals are different, it explains that in current paradigms of Pavlovian fear condition studies the times of paring of extinction stage is more than that of conditioning stage and retrieval stage (Goswami et al., 2010; Keller et al., 2015; Merz et al., 2018). The protocol we used in the present study (i.e., 7 CSs that end with US in the conditioning stage, 15 CSs in the extinction stage, and 6 CSs in the retrieval stage) has been proved effective and with good validity and was employed in our previous publications (Lin et al., 2020a; Lin et al., 2020b)
How long after the last behavioral test the animals were sacrificed? Please add this information in the method section.
Reply:The rats were sacrificed 24 hours after the ETM test, and the description of the timing of sacrifice has been addressed in the section of Methods and Materials and in the Schematic illustration of the experimental design (figure 1) in the revise version.
Which is the effect of the treatment in control animals? Even if from a translational point of view it is not useful to treat “healthy” subjects, how authors can compare and so comment the data obtained by comparing the SPS animals treated with CON animals untreated?
Reply: Instead of traditional 2x3 design, the present study employed one-way anova to reveal the statistical importance, and also following the suggestion from reviewer 2, Bonferroni post-hoc test was applied so it makes it easy to clarify (i) what is the difference between rats with SPS and rats without SPS, and once we clarify that, we analyze further that (ii) what is the difference among distinctive RU486 intervention timing. Thanks for your comment to point out that the statement regarding comparing the SPS animals treated with CON animals untreated was inappropriate, so in the revised version of the manuscript we removed the comparisons of SPS-RU486/Veh and SPS-Veh/RU486 with the CON-Veh/Veh in the revised version. And you may see that we resentenced the statistical method in a more clear, pertinent, and appropriate way in the section of Data analyses of the Materials and Methods “In the present study, one-way analysis of variance (ANOVA) with a between-subjects factor of group was performed on the data of the ETM test, western blot, and plasma corticosterone level. ……… In the present study, the statistical importance exhibited the difference between rats with SPS and rats without SPS, and the timing effects of RU486 intervention... “.
Why authors used the one way anova? They should use the two-way anova since the factors are the procedure and the treatment, so as consequence, all experimental groups should be exposed to both the variables. To state that the treatment has a real effect in SPS animals, authors should treat control rats and study the effect of the only treatment.
Reply: Thanks for the comment and yes two-way anova is the best method for the study design to investigate two determinnat factors, here, the effect of traumatic experience (i.e., SPS, two levels, with or without SPS) and the effect of RU486 interventoin timing (three levels, A, B, and C). As the restriction of animals alloted (due to complying to the 3R principles of animal study (our animal center is a AAALAC fully accredited one, thus the use of animal particularly the number of using is highly regulated), thus we need to reduce the animal number and refine the protocol of the study. Therefore, as we just mentioned, the present study employ a two-stage analysis for revealing the statistical importance, which makes it more easy to realize (i) what is the difference between rats with SPS and rats without SPS (stage 1), and once we clarify that, we analyze further that (ii) what is the difference among distinctive RU486 intervention timing. We stated this as a limitation in the final paragraph of the discussion section, in which we mentioned that the results obtained from our study were only suitable to interpret SPS rats (or patients diagnosed with PTSD), but not for the control rats (or healthy population).
For example, figure 5C: how authors can say that “early RU486 intervention after SPS increases FKBP5 (and also Egr-1, that is not significative p=0.059) in the mPFC”? The increase in FKBP5 in SPS-RU486/Veh could be due to the only treatment with RU486 (but it is impossible to evaluate this hypothesis if they don’t have CON-RU486 group). Indeed, the SPS-Veh/Veh and SPS-RU486/Veh are not statistically different ( p=0.054 for FKBP5 and p= 0.071 for Egr-1).
Reply: Thanks for pointing out this mistake, which has been corrected in the revised version. For better interpretation, we have defined the data with a p-value between 0.05-0.07 as trending in the revised version, and have stated this criterion in the data analysis paragraph of Methods and Materials section.
Another example in figure 4A: since the SPS-veh/veh group induced per se an increased expression of GR independently from the treatment, authors cannot compare SPS-RU486/Veh and SPS-Veh / RU486 with the CON/Veh/Veh since these experimental groups are characterized by different procedure (CON-SPS) and treatment (early-late).
Reply: We removed the comparisons of SPS-RU486/Veh and SPS-Veh/RU486 with the CON/Veh/Veh in the revised version.
In which subcellular fraction the protein levels have been measured? The subcellular fractionation and the levels of GR in the fraction, as well as the level of FKBP5 in the cytosol could lead information about GR translocation in the nucleus or its retention in the cytosol.
Reply: Total cellular proteins were measured in this study, and it has been stated in the methods and materials section of the revised version of manuscript.
Have the authors analyzed the whole hippocampus? The dissection of ventral and dorsal subregions could help the author to better interpret the effects found.
Reply: Whole hippocampus were measured in this study. We did not separate subregions of hippocampus to ventral and dorsal parts.
Authors should include the meaning of the results not in the figure legend but in the text. Please move the summary sentence at the end of each paragraph.
Reply: The meanings (and interpretations) of the results now were addressed in the main text, but not the legend of each figure.
Minor:
Please increase the quality of the figure 2. Some SEM are missing.
Reply: We re-draw the figure 2 in which SEMs were added.
In the supplementary, please add the experimental groups above the WB bands.
Reply: In the supplementary of the revised version of manuscript, we added the experimental groups and put them above the WB bands.
References:
Goswami, S., Cascardi, M., Rodriguez-Sierra, O.E., Duvarci, S. & Pare, D. (2010) Impact of predatory threat on fear extinction in Lewis rats. Learn Mem, 17, 494-501.
Keller, S.M., Schreiber, W.B., Stanfield, B.R. & Knox, D. (2015) Inhibiting corticosterone synthesis during fear memory formation exacerbates cued fear extinction memory deficits within the single prolonged stress model. Behavioural brain research, 287, 182-186.
Lin, C.C., Chen, T.Y., Cheng, P.Y. & Liu, Y.P. (2020a) Early life social experience affects adulthood fear extinction deficit and associated dopamine profile abnormalities in a rat model of PTSD. Progress in neuro-psychopharmacology & biological psychiatry, 101, 109914.
Lin, C.C., Cheng, P.Y. & Liu, Y.P. (2020b) Effects of early life social experience on fear extinction and related glucocorticoid profiles - behavioral and neurochemical approaches in a rat model of PTSD. Behavioural brain research, 391, 112686.
Merz, C.J., Hamacher-Dang, T.C., Stark, R., Wolf, O.T. & Hermann, A. (2018) Neural Underpinnings of Cortisol Effects on Fear Extinction. Neuropsychopharmacology : official publication of the American College of Neuropsychopharmacology, 43, 384-392.

Reviewer 2 Report
Although background of the study was written in a coherent manner, major deficiencies were found in the experimental methodology, statistical analysis, and presentation of results.
Authors indicated that n = 12 per group was used in the study, does this sample size apply to all the behavioral, western blot and stress hormone measurement? Reviewer suggests that inclusion of all individual data points into the bar graphs to provide more detailed information regarding the distribution of individual values and this will show the number of sample size too.
In statistical analysis, one-way ANOVA with Tukey post-hoc test for multiple comparisons was used to analyse the statistical group differences, reviewer suggests that if the data is normally distributed, the multiple comparisons should be analysed by Bonferroni post-hoc test, as it corrects for statistical type-II error.
Which results were analyzed by two-way ANOVA?
In the methodology and figure 2, it is unclear regarding which phase was represented by unconditioned stimulus. A representative diagram illustrating the experiments for figure 2 will be beneficial to readers on understanding the study design.
In figure 2, one would expect that the percentage of freezing is higher in SPS-Veh/Veh animals during the day 1 testing for conditioning and day 2 for extinction. However, the results showed no significant differences among groups, and this raised a doubt on the establishment of animal model using SPS.
The ETM test for measuring the degrees of conditioned and unconditioned anxiety (ETM avoidance 1 and escape phases) is rather confusing. The time-spent and immobility responses of the animals in both the enclosed and open-arms should be included in the result section. How can one rule out the possibility of habituation effects of animals and also the interruption phase when animals were placed at the end of the enclosed arm for two successive trials?
What are ETM test results of avoidance 2? Results for baseline, avoidance 1 and 2, and escape, as well as the trial data for ETM should be provided.
Why propylene glycol was used to dissolve RU486? Can it be dissolved in saline? When prolong used of propylene glycol and in high doses, it will lead to toxicity Lim et al 2014.
Why single dose of RU486 was selected to test for the behavioral study? Although authors have cited previous studies on the use of 20 mg per kg, these studies have never conducted any dose-response study on different dosages of RU486 (see Abdullahi et al. 2020 and Adamec et al. 2007)
References:
- Lim et al. Propylene Glycol Toxicity in Children. J Pediatr Pharmacol Ther. 2014 Oct-Dec; 19(4): 277–282. https://www.ncbi.nlm.nih.gov/pmc/articles/PMC4341412/
- Abdullahi et al. Protective effects of morphine in a rat model of post-traumatic stress disorder: Role of hypothalamic-pituitary-adrenal axis and beta- adrenergic system. Behavioural brain research 2020, 395, (112867), 19.
- Adamec et al. Involvement of noradrenergic and corticoid receptors in the consol-idation of the lasting anxiogenic effects of predator stress. Behavioural brain research 2007, 179, (2), 192-207.
Proofreading of the manuscript by a native English speaker is highly recommended. Please check throughout the manuscript for grammatical errors. For examples:
Pg. 11: “All rats were housed together (2 rats/cage) at the same temperature, in a humidity-controlled holding facility (25°C ± 1°C) with 12-h light–dark cycles (lights on from 07:00–19:00) and received a standard laboratory chow diet (Ralston Purina, St. Louis, MO, USA) and sterile water ad libitum.”
>>> All rats (2 rats/cage) were housed in holding facility with controlled temperature (25°C ± 1°C), humidity (??), and 12-h light-dark cycles (lights on from 07:00–19:00). All animals received standard laboratory chow diet (Ralston Purina, St. Louis, MO, USA) and sterile water ad libitum.
Pg. 11: “All the experiments were assessed and approved by the NDMC ……”
>>> All experiments were conducted with approval from the NDMC ……
Pg. 12: “In the end of experiment (24 h after the ETM test), the rats were placed ……”
>>> At the end of experiment (24 h after the ETM test), all rats were placed ……
Pg. 12: “After 24 h, the rats were placed at the end of the enclosed arm and the time required to leave the arm by using all four paws was be recorded and used as the baseline latency.”
>>> Pg. 12: After 24 h, the rats were placed at the end of the enclosed arm and the time required to leave the arm by using all four paws was recorded and used as the baseline latency.
Author Response
Dear reviewer 2, April 30, 2022
Thanks for your comments on IJMS-1663969, and we realize that a lot of efforts you have made to make this manuscript more readable and concise, so it may provide greater scientific contribution to the research field of fear memory. Following your suggestions we revised the manuscript fully and also sent it for a carefully English Editing work from MDPI before its submission. Please see our replies to your comment point by point as the following.
Best Regards,
Liu, Yia-Ping M.D., Ph.D
Professor, Department of psychiatry, Cheng Hsin General Hospital, Taipei, Taiwan Director, Laboratory of cognitive neuroscience, Department of physiology, National Defense Medical Center, Taipei, Taiwan
Although background of the study was written in a coherent manner, major deficiencies were found in the experimental methodology, statistical analysis, and presentation of results.
Authors indicated that n = 12 per group was used in the study, does this sample size apply to all the behavioral, western blot and stress hormone measurement?
Reply: All 12 rats in each group will be used for behavioral tests. 6 and 8 rats from each group will be randomly selected for the western blot and plasma corticosterone level assay, respectively. We added this statement in the section of methods and materials and also stated the number of rats employed to each experiment in the legend of corresponding figure.
Reviewer suggests that inclusion of all individual data points into the bar graphs to provide more detailed information regarding the distribution of individual values and this will show the number of sample size too.
Reply: Thanks for this suggestion, it makes the graph more understandable. We added all individual data points in their corresponding bar graph in the revise version of the manuscript.
In statistical analysis, one-way ANOVA with Tukey post-hoc test for multiple comparisons was used to analyse the statistical group differences, reviewer suggests that if the data is normally distributed, the multiple comparisons should be analysed by Bonferroni post-hoc test, as it corrects for statistical type-II error.
Reply: We followed your suggestion and in the revised version of the manuscript all multiple comparisons used in this study has been changed to Bonferroni post-hoc test.
Which results were analyzed by two-way ANOVA?
Reply: Two-way ANOVA (for group, the between-subject factor, and for trial, the within-subject factor) were performed to reveal the effects of group and trial on the freezing level of the fear conditioning test (seeing figure 2). Statistical analysis has been rewritten in the section of methods and materials in the revised version of the manuscript.
In the methodology and figure 2, it is unclear regarding which phase was represented by unconditioned stimulus. A representative diagram illustrating the experiments for figure 2 will be beneficial to readers on understanding the study design.
Reply: In the revised version, we redrew the figure 2 in which representative diagrams were placed corresponding to Day1, Day2, and Day3. This can make it more understandable about phase, conditioned stimulus, and unconditioned stimulus.
In figure 2, one would expect that the percentage of freezing is higher in SPS-Veh/Veh animals during the day 1 testing for conditioning and day 2 for extinction. However, the results showed no significant differences among groups, and this raised a doubt on the establishment of animal model using SPS.
Reply: It is still inconclusive for the effects on fear conditioning (i.e., Day 1) and fear extinction learning (i.e., Day 2), as both increase in freezing and remaining unchanged reported (Knox et al., 2012a; Knox et al., 2012b; George et al., 2015; Chaby et al., 2020). However, it is more conclusive that the freezing in fear extinction retrieval (i.e., Day 3) shows increased in SPS animals. In other words, the extinction retrieval is more sensitive than condition and extinction in rats with SPS experience. In this regard, our data in the present study was consistent with Konx et al. (2012), Pitman et al. (2012), George et al. (2015), and Fang et al. (2018) that SPS impairs fear extinction retrieval but not fear conditioning and fear extinction (Knox et al., 2012a; Pitman et al., 2012; George et al., 2015; Lin et al., 2018).
The ETM test for measuring the degrees of conditioned and unconditioned anxiety (ETM avoidance 1 and escape phases) is rather confusing. The time-spent and immobility responses of the animals in both the enclosed and open-arms should be included in the result section. How can one rule out the possibility of habituation effects of animals and also the interruption phase when animals were placed at the end of the enclosed arm for two successive trials?
Reply: There were 4 trials in a session of ETM: baseline, avoidance latency 1, avoidance latency 2, and escape latency. For baseline, avoidance latency 1, and avoidance latency 2, rats are put into the enclosed arm and the time spent from rats being released of experimenter’s hand to rats leaving the enclosed arm is recorded (that is why it mimics the conditioned anxiety disorder as it is a process of leaving the safe place for three times), whereas for escape latency, rats are put into the opened arm and the time spent from rats being released from experimenter’s hand to rats leaving the opened arm is recorded (that is why it mimics the unconditioned anxiety as rats are placed in a quite appalling situation just for one time and the nature of rats are trying to escape from the situation as soon as possible if it is stressful, thus for example, rats with high anxiety level are with short time span in the opened arm). It is therefore one of the advantages of ETM is to distinguish generalized anxiety and panic anxiety (Poltronieri et al., 2003; Campos et al., 2013).
I think you mentioned an important point is that the possibility of habituation may occur, particularly during the first three trials (i.e., baseline, avoidance latency 1, and avoidance latency 2) as rats are put back to the homecage and placed into the enclosed arm again. To avoid this possibility, once the rats exited the enclosed arm (baseline and avoidance latency 1 and 2) the trial was stopped and the rat was placed into their homecage immediately to inhibit the habituation effects (and that is also why this kind design of ETM study, latency is superior than the total time rats spent in the enclosed arm, for the latter rats may leave and re-entry the enclosed arm many times). However, the habituation is still unavoidable, as you can see in the present study the data of avoidance 2 latency showed no significant difference among the groups which is possibly due to the ceiling effect caused by the habituation, and that is why avoidance latency 1 rather than avoidance latency 2 is normally chosen for indexing rats’ anxiety level. Note the baseline latency is also not appropriate for this use as some researchers consider the drive of fleeing from experimenter’s hand can also be a confounding variable (Zangrossi & Graeff, 1997; 2014). And that comes the current design to employ avoidance latency 1 to reveal rats’ anxiety.
What are ETM test results of avoidance 2? Results for baseline, avoidance 1 and 2, and escape, as well as the trial data for ETM should be provided.
Reply: We provide the results for baseline, avoidance 1 and 2, and escape (figure 3) in the revised version of the manuscript.
Why propylene glycol was used to dissolve RU486? Can it be dissolved in saline? When prolong used of propylene glycol and in high doses, it will lead to toxicity Lim et al 2014.
Reply: RU486 cannot be dissolved in saline. RU486 was dissolved in rat model of studies in propylene glycol for systemic injection (Dong et al., 2017; Frank et al., 2019). In order to exclude the potential effects of propylene glycol, the CON-Veh/Veh and SPS-Veh/Veh groups were also given propylene glycol for 14 days.
Why single dose of RU486 was selected to test for the behavioral study? Although authors have cited previous studies on the use of 20 mg per kg, these studies have never conducted any dose-response study on different dosages of RU486 (see Abdullahi et al. 2020 and Adamec et al. 2007)
Reply: Due to the restriction of rat number used in the present study, there was a lack of dose-response data of RU486 on fear and anxiety behaviors. Thus it should cautious to extend the current result to the full range of dose of RU486. We address this as a limitation at the end of discussion section in the revised version of manuscript.
References:
- Lim et al. Propylene Glycol Toxicity in Children. J Pediatr Pharmacol Ther. 2014 Oct-Dec; 19(4): 277–282. https://www.ncbi.nlm.nih.gov/pmc/articles/PMC4341412/
- Abdullahi et al. Protective effects of morphine in a rat model of post-traumatic stress disorder: Role of hypothalamic-pituitary-adrenal axis and beta- adrenergic system. Behavioural brain research 2020, 395, (112867), 19.
- Adamec et al. Involvement of noradrenergic and corticoid receptors in the consol-idation of the lasting anxiogenic effects of predator stress. Behavioural brain research 2007, 179, (2), 192-207.
Proofreading of the manuscript by a native English speaker is highly recommended. Please check throughout the manuscript for grammatical errors.
Reply: The revised manuscript has been sent to English Editing work from MDPI Editing Service before it submission.
For examples:
Pg. 11: “All rats were housed together (2 rats/cage) at the same temperature, in a humidity-controlled holding facility (25°C ± 1°C) with 12-h light–dark cycles (lights on from 07:00–19:00) and received a standard laboratory chow diet (Ralston Purina, St. Louis, MO, USA) and sterile water ad libitum.”
>>> All rats (2 rats/cage) were housed in holding facility with controlled temperature (25°C ± 1°C), humidity (??), and 12-h light-dark cycles (lights on from 07:00–19:00). All animals received standard laboratory chow diet (Ralston Purina, St. Louis, MO, USA) and sterile water ad libitum.
Reply: It has been revised as suggested.
Pg. 11: “All the experiments were assessed and approved by the NDMC ……”
>>> All experiments were conducted with approval from the NDMC ……
Reply: It has been revised as suggested.
Pg. 12: “In the end of experiment (24 h after the ETM test), the rats were placed ……”
>>> At the end of experiment (24 h after the ETM test), all rats were placed ……
Reply: It has been revised as suggested.
Pg. 12: “After 24 h, the rats were placed at the end of the enclosed arm and the time required to leave the arm by using all four paws was be recorded and used as the baseline latency.”
>>> Pg. 12: After 24 h, the rats were placed at the end of the enclosed arm and the time required to leave the arm by using all four paws was recorded and used as the baseline latency.
Reply: It has been revised as suggested.
References:
Campos, A.C., Fogaca, M.V., Aguiar, D.C. & Guimaraes, F.S. (2013) Animal models of anxiety disorders and stress. Braz J Psychiatry, 35 Suppl 2, S101-111.
Chaby, L.E., Sadik, N., Burson, N.A., Lloyd, S., O'Donnel, K., Winters, J., Conti, A.C., Liberzon, I. & Perrine, S.A. (2020) Repeated stress exposure in mid-adolescence attenuates behavioral, noradrenergic, and epigenetic effects of trauma-like stress in early adult male rats. Scientific reports, 10, 020-74481.
Dong, L., Wang, S., Li, Y., Zhao, Z., Shen, Y., Liu, L., Xu, G., Ma, C., Li, S., Zhang, X. & Cong, B. (2017) RU486 Reverses Emotional Disorders by Influencing Astrocytes and Endoplasmic Reticulum Stress in Chronic Restraint Stress Challenged Rats. Cellular physiology and biochemistry : international journal of experimental cellular physiology, biochemistry, and pharmacology, 42, 1098-1108.
Frank, M.G., Annis, J.L., Watkins, L.R. & Maier, S.F. (2019) Glucocorticoids mediate stress induction of the alarmin HMGB1 and reduction of the microglia checkpoint receptor CD200R1 in limbic brain structures. Brain, behavior, and immunity, 80, 678-687.
George, S.A., Rodriguez-Santiago, M., Riley, J., Rodriguez, E. & Liberzon, I. (2015) The effect of chronic phenytoin administration on single prolonged stress induced extinction retention deficits and glucocorticoid upregulation in the rat medial prefrontal cortex. Psychopharmacology, 232, 47-56.
Knox, D., George, S.A., Fitzpatrick, C.J., Rabinak, C.A., Maren, S. & Liberzon, I. (2012a) Single prolonged stress disrupts retention of extinguished fear in rats. Learn Mem, 19, 43-49.
Knox, D., Nault, T., Henderson, C. & Liberzon, I. (2012b) Glucocorticoid receptors and extinction retention deficits in the single prolonged stress model. Neuroscience, 223, 163-173.
Lin, C.-S., Wu, C.-Y., Wu, S.-Y. & Lin, H.-H. (2018) Brain activations associated with fearful experience show common and distinct patterns between younger and older adults in the hippocampus and the amygdala. Scientific reports, 8, 5137.
Pitman, R.K., Rasmusson, A.M., Koenen, K.C., Shin, L.M., Orr, S.P., Gilbertson, M.W., Milad, M.R. & Liberzon, I. (2012) Biological studies of post-traumatic stress disorder. Nature reviews. Neuroscience, 13, 769-787.
Poltronieri, S.C., Zangrossi, H. & de Barros Viana, M. (2003) Antipanic-like effect of serotonin reuptake inhibitors in the elevated T-maze. Behavioural brain research, 147, 185-192.
Zangrossi, H., Jr. & Graeff, F.G. (1997) Behavioral validation of the elevated T-maze, a new animal model of anxiety. Brain research bulletin, 44, 1-5.
Zangrossi, H., Jr. & Graeff, F.G. (2014) Serotonin in anxiety and panic: contributions of the elevated T-maze. Neuroscience and biobehavioral reviews, 46 Pt 3, 397-406.

Round 2
Reviewer 1 Report
Minor issues remain:
Since authors consider the whole hippocampus instead of the ventral and dorsal subregions, known to have different functions, they should add this as limitation since some effect they found (or did not find) could be due also to this aspect.
In the limitation, please replace “healthy individuals” with control animals in the sentence: “Third, the present study did not investigate the effect of RU486 on healthy individuals”.
Authors should add in the figure legends the statistics employed.
In the paragraph, 4.8, please modify the reference 74 (now is indicated as 7474).
In the 4.9 paragraph, the sentence “In the present study, the statistical importance exhibited the difference between rats with SPS and rats without SPS, and the timing effects of RU486 intervention” is not clear and not appropriate for this section. Please rephrase the sentence indicating the factors employed for the two-way analysis (SPS and timing of treatment).
Author Response
Dear Reviewer 1, May 6, 2022
Appreciate so much for your reviewing of the manuscript IJMS-1663969 again. We appreciate you for your many extraordinary efforts to help us improve the manuscript. Following your suggestions we revised the manuscript carefully. Please see our replies to your comments (In bold type).
Best Regards,
Liu, Yia-Ping M.D., Ph.D
Professor, Department of psychiatry, Cheng Hsin General Hospital, Taipei, Taiwan Director, Laboratory of cognitive neuroscience, Department of physiology, National Defense Medical Center, Taipei, Taiwan
Minor issues remain:
Since authors consider the whole hippocampus instead of the ventral and dorsal subregions, known to have different functions, they should add this as limitation since some effect they found (or did not find) could be due also to this aspect.
Reply: Thanks for the comment and we added this in the limitation in the section of discussion “Forth, given that ventral and dorsal subregions of hippocampus have different functions [58], the use of whole hippocampus in the present study may make some potential effects that cannot be found.”
In the limitation, please replace “healthy individuals” with control animals in the sentence: “Third, the present study did not investigate the effect of RU486 on healthy individuals”.
Reply: We replaced the “healthy individuals” with control animals in the sentence in the revised version.
Authors should add in the figure legends the statistics employed.
Reply: We added the statistics employed in the figure legends in the revised version.
In the paragraph, 4.8, please modify the reference 74 (now is indicated as 7474).
Reply: Thanks for pointing out this mistake, which has been corrected in the revised version.
In the 4.9 paragraph, the sentence “In the present study, the statistical importance exhibited the difference between rats with SPS and rats without SPS, and the timing effects of RU486 intervention” is not clear and not appropriate for this section. Please rephrase the sentence indicating the factors employed for the two-way analysis (SPS and timing of treatment).
Reply: Thanks for your comment. For the description of the analysis method of the present study, one way ANOVA with four groups of manipulations was employed, in which the effects of different intervention timing of RU486 were examined in SPS rats.

Reviewer 2 Report
Authors have addressed all concerns raised by me.
Author Response
Dear Reviewer 2, May 6, 2022
Appreciate so much for your reviewing of the manuscript IJMS-1663969 again. We thank you for your many extraordinary efforts to help us improve the manuscript.
Best Regards,
Liu, Yia-Ping M.D., Ph.D
Professor, Department of psychiatry, Cheng Hsin General Hospital, Taipei, Taiwan Director, Laboratory of cognitive neuroscience, Department of physiology, National Defense Medical Center, Taipei, Taiwan
Comments and Suggestions for Authors
Authors have addressed all concerns raised by me.
